# Wave-Mamba: Wavelet State Space Model for Ultra-High-Definition Low-Light Image Enhancement

Wenbin Zou
South China University of Technology
PAZHOU LAB
Guangzhou, China
alexzou14@foxmail.com

Hongxia Gao*
South China University of Technology
PAZHOU LAB
Guangzhou, China
hxgao@scut.edu.cn

Weipeng Yang
South China University of Technology
Guangzhou, China
wpyscut@foxmail.com

Tongtong Liu
South China University of Technology
Guangzhou, China
202310183310@mail.scut.edu.cn

## ABSTRACT

Ultra-high-definition (UHD) technology has attracted widespread attention due to its exceptional visual quality, but it also poses new challenges for low-light image enhancement (LLIE) techniques. UHD images inherently possess high computational complexity, leading existing UHD LLIE methods to employ high-magnification downsampling to reduce computational costs, which in turn results in information loss. The wavelet transform not only allows downsampling without loss of information, but also separates the image content from the noise. It enables state space models (SSMs) to avoid being affected by noise when modeling long sequences, thus making full use of the long-sequence modeling capability of SSMs. On this basis, we propose Wave-Mamba, a novel approach based on two pivotal insights derived from the wavelet domain: 1) most of the content information of an image exists in the low-frequency component, less in the high-frequency component. 2) The high-frequency component exerts a minimal influence on the outcomes of low-light enhancement. Specifically, to efficiently model global content information on UHD images, we proposed a low-frequency state space block (LFSSBlock) by improving SSMs to focus on restoring the information of low-frequency sub-bands. Moreover, we propose a high-frequency enhance block (HFEBlock) for high-frequency sub-band information, which uses the enhanced low-frequency information to correct the high-frequency information and effectively restore the correct high-frequency details. Through comprehensive evaluation, our method has demonstrated superior performance, significantly outshining current leading techniques while maintaining a more streamlined architecture. The code is available at https://github.com/AlexZou14/Wave-Mamba.

*The corresponding author.

## CCS CONCEPTS

• **Computing methodologies** → **Image processing**.

## KEYWORDS

Low-light image enhancement, State Space Model, Ultra-High-Definition

**ACM Reference Format:**
Wenbin Zou, Hongxia Gao, Weipeng Yang, and Tongtong Liu. 2024. Wave-Mamba: Wavelet State Space Model for Ultra-High-Definition Low-Light Image Enhancement. In *Proceedings of the 32nd ACM International Conference on Multimedia (MM '24), October 28-November 1, 2024, Melbourne, VIC, AustraliaProceedings of the 32nd ACM International Conference on Multimedia (MM'24), October 28-November 1, 2024, Melbourne, Australia.* ACM, New York, NY, USA, 10 pages. https://doi.org/10.1145/3664647.3681580

## 1 INTRODUCTION

The rapid advancements in imaging technology have enabled the widespread adoption of Ultra-High Definition (UHD) across diverse applications. However, the increased pixel count and high-resolution nature of UHD images introduce significant challenges. UHD images are more susceptible to noise and lighting effects during capture, degrading quality and impacting high-level vision tasks. In this work, we focus on the crucial task of low-light image enhancement (LLIE) for UHD images.

With the significant success of Convolutional Neural Networks (CNNs) and Transformers [46, 48, 49] in the field of image restoration, a lot of learning-based methods [2, 28, 32, 38, 47] have been proposed to tackle the task of LLIE. Although these methods have achieved remarkable performance on existing low-light datasets, they are trained on the LOL [28] and SID [3] datasets, where all images have resolutions below 1K (1920×1080). Furthermore, existing methods have sought higher performance through the design of more complex networks and an increase in network parameters. However, due to the discrepancy between the data distribution of UHD images and that of existing datasets, these advanced methods are not effectively applicable to 4K (3840×2160) scenarios. Therefore, as UHD images become increasingly prevalent, the domain of image restoration for UHD images is garnering more attention from researchers.

With the release of some UHD LLIE datasets, such as UHD-LOL [13] and UHD-LL [15], many methods [13, 15, 23, 31] tailored for

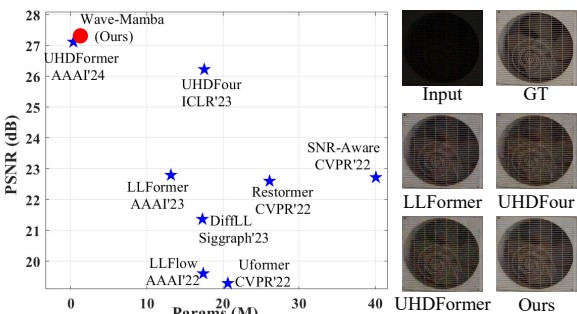

**Figure 1: Model parameters and performance comparison and visual comparisons.**

UHD LLIE have also been proposed. Among these, Wang et al. [13] introduced an end-to-end UHD LLIE framework by incorporating Transformers and UNet. Leveraging the exceptional ability of the Transformer to capture long-distance information, the proposed method achieved superior restoration performance. However, due to the high computational cost of Transformers, this method could not be efficiently implemented for full-resolution UHD image inference on edge devices. To enable full-resolution inference of UHD images on consumer-grade GPUs, Li et al. [15] reduced the resolution of UHD images by 8× and embedded Fourier transform into a cascaded network, thus obtaining satisfactory results on real datasets. On this basis, Wang et al. [23] proposed a correction-matching Transformer module that utilizes high-resolution information to correct low-resolution features, achieving impressive performance through parallel enhancement of high and low-resolution. While these methods have demonstrated remarkable performance, they rely on significant downsampling to reduce computational costs, inevitably leading to the loss of critical image information. This process undoubtedly compromises the quality of image restoration. *Therefore, enhancing the ability of the network to augment global information without sacrificing image detail is crucial.*

Inspired by the property of state space models (SSMs) [6, 8, 22] that long sequences can be modeled using linear complexity, it makes it possible to model global information in UHD images. In particular, the improved Mamba [6] has been successfully applied to several computer vision tasks [18, 20, 41] and achieved significant performance. However, the unidirectional modeling property of SSMs makes it susceptible to noise, which hinders the application of mamba to low-light scenes containing complex noise. *Therefore, how to effectively apply the long sequence modeling capability of SSMs to UHD LLIE is a question worth exploring.*

To address the challenges of UHD low-light imaging, we propose an efficient method, called Wave-Mamba, that combines wavelet transform with Mamba. Unlike existing UHD LLIE methods, our approach avoids traditional downsampling, instead employing wavelet transform to prevent information loss. Additionally, the wavelet transform separates image content from noise, overcoming the limitations of standard SSMs, which are insensitive to noise. Particularly, our method is designed based on two observations in the wavelet domain of low-light noisy images: *1. In the wavelet domain, most image information resides in the low-frequency component, with only a minor portion of texture information in the high-frequency component. 2. High-frequency information has a*

*minimal impact on the results of LLIE.* Inspired by these insights, our network design focuses on processing low-frequency information, using enhanced low-frequency information to adjust the high-frequency information, effectively saving computational resources. Specifically, we developed a Low-Frequency State Space Block (LFSSBlock) that leverages the robust global modeling capability of SSMs to effectively enhance the illumination and texture information within the low-frequency component. For the high-frequency component, we designed a High-Frequency Enhance Block (HFEBlock), which utilizes the enhanced low-frequency information to match and correct the high-frequency data, achieving accurate and clear textures. With the aforementioned design, our Wave-Mamba significantly reduces computational costs while delivering outstanding performance, as illustrated in Figure 1.

Our key contributions are summarized as follows:

- We are the pioneers in introducing Mamba to the UHD LLIE task, proposing a novel method named Wave-Mamba, inspired by unique characteristics observed in the wavelet domain. Unlike existing UHD LLIE methods, our method exploits the wavelet transform to avoid information loss, and to overcome the shortcomings of SSMs which are insensitive to noise.
- We propose a Low-Frequency State Space Block (LFSSBlock) that leverages the linear complexity and powerful information modeling capabilities of State Space Models for enhancement. This effectively balances performance with computational costs.
- We propose a High-Frequency Enhance Block (HFEBlock) that employs enhanced low-frequency information for match correction, thereby effectively avoiding texture errors and loss.
- The proposed Wave-Mamba exhibits extraordinary effectiveness and efficiency in addressing the combined tasks of illumination enhancement and noise removal in ultra-high-definition images.

## 2 RELATED WORK

### 2.1 Low-light Image Enhancement

It has seen substantial developments with the advent of various innovative models and frameworks aimed at improving underexposed photos and videos. Wang et al. [25] introduced networks for image-to-illumination mapping, while Zero-DCE [10], and its extension Zero-DCE++ [14] have made significant strides in enhancing image brightness and visual appeal through image-specific curve estimation trained with non-reference losses. Moreover, Wang et al. [5] presented a dual-stage low-light image enhancement network, FourLLIE, which enhances brightness by estimating amplitude transformation mappings in the frequency domain. Feng et al. [24] proposed a learnability enhancement strategy based on noise modeling, which improves the denoising performance of raw images in low-light conditions. Additionally, diffusion models, as seen in DiffLL [12], utilize a sequence of denoising refinements for realistic detail generation in LLIE tasks, showing potential for low-light image enhancement. Despite these methods achieving excellent results, the direct application to Ultra-High-Definition (UHD) images is constrained by the high computational demand,

marking a challenge for future research directions in efficiently processing UHD content.

## 2.2 UHD Image Restoration

In recent years, UHD image restoration has emerged as a field of growing interest, with significant contributions from researchers [9, 16, 34, 45]. Zheng et al. [39, 40] pioneered the use of bilateral learning for UHD image dehazing and High Dynamic Range (HDR), leveraging the concept of learning local affine coefficients from downscaled images to enhance images at their original resolution. Innovations such as UHD-SFNet [29] and FourUHD [15] have advanced the field by focusing on enhancing underwater and low-light UHD images within the Fourier domain, capitalizing on the insight that illumination primarily resides in amplitude components.

To address the limitations imposed by the need for downsampling in traditional UHD restoration techniques, NSEN [33] introduced an innovative, spatially-variant, and reversible downsampling method. This approach dynamically adapts the downsampling rate to the image's detail density, enhancing the detail preservation in the restoration process. Furthermore, LLFormer [13] represents the first attempt to employ a transformer-based model for the UHD Low-Light Image Enhancement (UHD-LLIE) task. Despite its pioneering status, LLFormer encounters challenges in executing full-resolution inference on standard consumer GPUs.

## 2.3 State Space Models (SSMs)

Derived from control theory fundamentals, SSMs [7, 8, 22] have advanced remarkably in deep learning, exhibiting extraordinary efficiency in handling long-range dependencies due to their linear scalability with sequence length. Recently, the emergence of Mamba [6], a selective, data-focused SSM optimized for hardware, has outperformed Transformer models in NLP tasks, displaying linear scalability concerning input size. Furthermore, the application of Mamba has extended to vision-related tasks, including image classification [18, 41], image restoration [11, 21], and biomedical image segmentation [20]. On this basis, some researchers [42–44] combine linear representation with incremental learning to propose efficient incremental learning methods. Currently, while Mamba-based image restoration approaches show promise, they fall short of direct application to UHD image restoration due to inference challenges. Given the exceptional computational efficiency and scalability of Mamba, this study aims to pioneer the application of Mamba in UHD low-light image enhancement.

## 3 PRELIMINARIES: STATE SPACE MODELS

Structured State Space Models (S4), fundamentally based on continuous systems, explain the dynamic relationship between inputs $x(t)$ and outputs $y(t)$ within linear time-invariant frameworks. Essentially, this system maps a one-dimensional function or sequence $x(t) \in \mathbb{R}^L$ to $y(t) \in \mathbb{R}^L$ via an implicit latent state $h(t) \in \mathbb{R}^N$. From a mathematical perspective, this system is succinctly represented by a linear ordinary differential equation (ODE), detailed as follows:

$$h'(t) = Ah(t) + Bx(t) \tag{1}$$

$$y(t) = Ch(t) + Dx(t) \tag{2}$$

where $A \in \mathbb{R}^{N \times N}$, $B \in \mathbb{R}^{N \times 1}$, $C \in \mathbb{R}^{1 \times N}$ are the parameters for a state size $N$, and $D \in \mathbb{R}^1$ denotes the skip connection.

To integrate SSMs within deep learning algorithms, researchers have discretized the aforementioned ODE process and aligned the model with the sample rate of the underlying signal present in the input data. Typically, this discretization employs the zeroth-order hold (ZOH) method, incorporating the time scale parameter $\Delta$ to transition continuous parameters $A$ and $B$ into their discrete counterparts $\bar{A}$ and $\bar{B}$. This process is defined as follows:

$$h'_t = \bar{A}h_{t-1} + \bar{B}x_t \tag{3}$$

$$y_t = Ch_t + Dx_t \tag{4}$$

$$\bar{A} = e^{\Delta A} \tag{5}$$

$$\bar{B} = (\Delta A)^{-1}(e^{\Delta A} - I) \cdot \Delta B \tag{6}$$

where $\Delta \in \mathbb{R}^D$ and $B, C \in \mathbb{R}^{D \times N}$.

The recently developed state space model, Mamba [19], has been further improved to make the parameters $B$, $C$, and $\Delta$ dependent on the input, thereby enabling dynamic feature representation. In essence, Mamba adopts a similar recursive structure as seen in Eq. (3), allowing it to process and retain information from exceptionally long sequences. This capability ensures that a greater number of pixels contribute to the restoration process. Additionally, Mamba utilizes a parallel scan algorithm [19], mirroring the parallel processing benefits outlined in Eq. (3), thereby optimizing the training and inference process for efficiency.

## 4 METHODOLOGY

In this section, we first discuss the observation and method design motivation when analyzing low-light images in the wavelet domain. Next, the structure of our proposed method and the proposed modules were described in detail.

### 4.1 Observations in Wavelet Domain

Here, we provide additional discussion details on low-light images in the wavelet domain to clarify our observations emphasized in the Sec. 1, and provide a concise explanation in Figure 3.

**Wavelet Transformation**: Given an image $I \in \mathbb{R}^{H \times W \times C}$, it is decomposed into four frequency subbands by wavelet transform.

$$\{cA, cH, cV, cD\} = DWT(I) \tag{7}$$

where $cA, cH, cH, cD \in \mathbb{R}^{\frac{H}{2} \times \frac{W}{2} \times C}$ denote the low-frequency information of the input image and the high-frequency information in three different directions (vertical, horizontal, and diagonal directions), respectively. The $DWT(\cdot)$ denotes the 2D discrete wavelet transform operation. Subsequently, we can reconstruct the decomposed frequency subbands to the original map without any loss of information by inverse wavelet transform $IWT(\cdot)$, i.e.

$$I = IWT(cA, cH, cV, cD) \tag{8}$$

Therefore, leveraging this characteristic enables the execution of multiple DWT on an image, effectively downsampling the image while preserving information. This is one of the main reasons that prompted us to use DWT for the UHD LLIE task.

**Observations I:** As illustrated Figure 3, we decompose both low-light and normal-light images using a 2D wavelet transform. Subsequently, the decomposed four frequency sub-bands are broadly

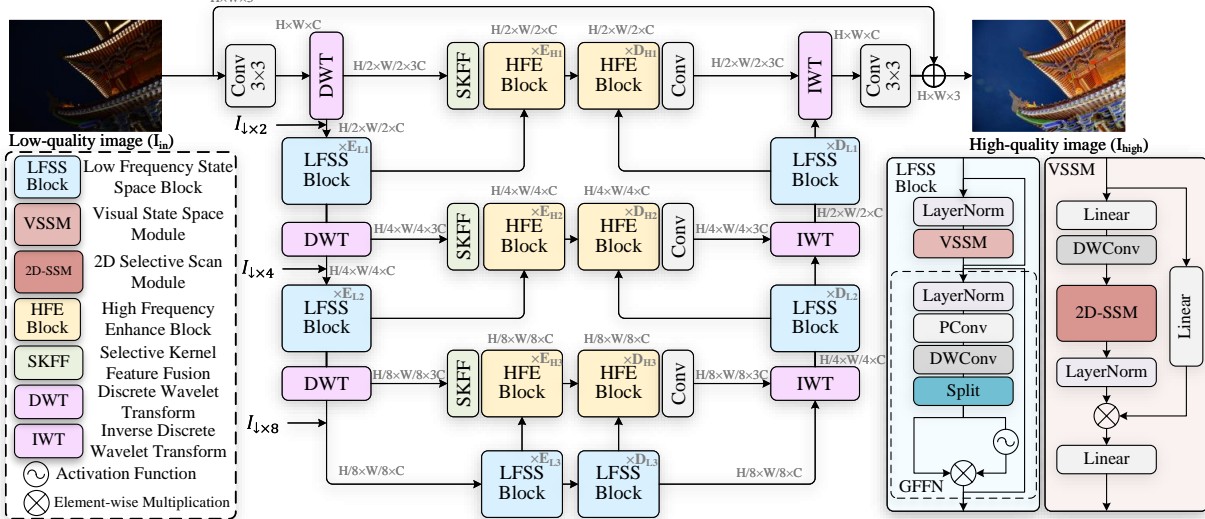

**Figure 2: The overall architecture of the proposed Wave-Mamba. The Wave-Mamba performs up and down sampling by utilizing the wavelet transform, and feature extraction and enhancement by using Low-Frequency State Space Block (LFSSBlock) and High-Frequency Enhance Block (HFEBlock) for the low-frequency and high-frequency components, respectively.**

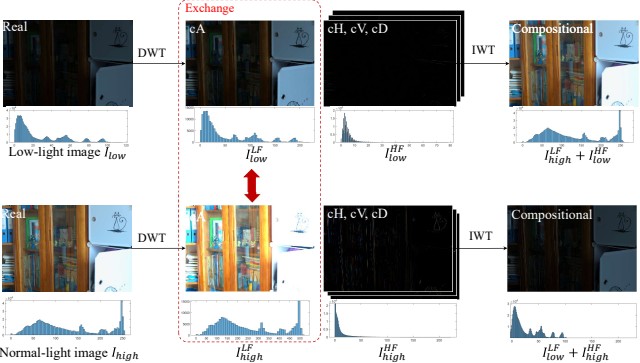

**Figure 3: Observations in Wavelet Domain. As can be seen from the figure, by exchanging the low-frequency components of low and normal lighting image content changes dramatically, while changing the high-frequency components does not effectively improve image quality.**

categorized into low and high frequencies, and histograms for the original image $I_{low}$, $I_{high}$, low-frequency component $I_{low}^{LF}$, $I_{high}^{LF}$, and high-frequency component $I_{low}^{HF}$, $I_{high}^{HF}$ are generated. From the diagrams, it is evident that for both normal and low-light images, the histogram of the low-frequency information more closely resembles the histogram distribution of the input image. Additionally, visualizations reveal that the information contained within the high-frequency component is significantly less than that in the low-frequency component. Hence, we can easily conclude: ***In the wavelet domain, the majority of image information is found in the low-frequency component, while a smaller fraction of texture details resides in the high-frequency component.***

**Observations II:** We further experimented with the reconstruction process during the inverse wavelet transformation. Specifically, we reconstructed the images by swapping the corresponding low-frequency information and then generated the respective histograms. As illustrated in Figure 3, images reconstructed with

different high-frequency sub-bands still maintain a histogram distribution similar to that of the original image, whereas those reconstructed with different low-frequency sub-bands exhibit a significant alteration in the image's histogram distribution. From a visual perspective, the content of the image is highly correlated with the reconstructed low-frequency component. ***Therefore, compared to the low-frequency sub-bands, the high-frequency sub-bands have a lesser impact on LLIE.***

## 4.2 Framework Overview

The above observations and analyses of low-light images in the wavelet domain have inspired us to design an efficient framework using wavelet transform for the UHD LLIE task, named Wave-Mamba, as shown in Figure 2. Specifically, we have introduced an effective Mamba architecture to create a Low-Frequency State Space Block (LFSSBlock), targeting the information-rich low-frequency component. This enables the network to enhance and repair global information with linear complexity. Additionally, we propose a High-Frequency Enhance Block (HFEBlock) that utilizes the enhanced low-frequency component to correct high-frequency information, ensuring the network accurately captures high-frequency texture information. We provide the overall pipeline of our method and further details on the critical components of our approach below.

Figure 2 shows our overall framework of Wave-Mamba. Our Wave-Mamba is constructed using a multi-scale UNet structure. In the downsampling phase, we employ DWT to prevent the information loss encountered with traditional downsampling operations. In line with the observations made in Sec. 4.1, we specifically apply DWT downsampling to the low-frequency component, which contains a greater amount of information, thus effectively reducing computational costs. Regarding the high-frequency component, we employ the enhanced low-frequency component for the correction and restoration of high-frequency information.

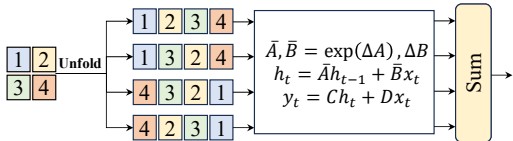

**Figure 4: The architecture of the 2D selective scanning module (2D-SSM) in the VSS Module.**

Specifically, given UHD low-light image $I_{in} \in \mathbb{R}^{H \times W \times 3}$, we first apply a 3×3 convolution to obtain low-level embeddings $F_0 \in \mathbb{R}^{H \times W \times C}$, where $H$, $W$, and $C$ denotes height, width, and channel, respectively. Then the hierarchical encoding is achieved through three layers of DWT downsampling and LFSS blocks. The low-frequency feature $F_{L1}$, $F_{L2}$ and $F_{L3}$, processed by DWT downsampling at different layers, are downsampled to size $\frac{H}{2} \times \frac{W}{2}$, $\frac{H}{4} \times \frac{W}{4}$, and $\frac{H}{8} \times \frac{W}{8}$ respectively. To fully utilize the input image information, we also fused obtained $i$-th layer of low-frequency feature $F_{Li}$ with the corresponding downsampled input image $I_{\downarrow \times 2^i}$ and the enhanced low-frequency features $F_{Li}^e$ are obtained by adjusting the global information through the stack LFSS blocks. The high-frequency features $F_{H1}$, $F_{H2}$, and $F_{H3}$ of each layer are first reduced from channel $3 * C$ to channel $C$ by Selective Kernel Feature Fusion (SKFF) [36], thus facilitating correction by the HFEBlock using the low-frequency components. Subsequently, the enhanced high-frequency features undergo further refinement in conjunction with the enhanced low-frequency features, employing the LFSSBlock and the HFEBlock, respectively. The restored features are progressively upsampled using the wavelet inverse transform through each layer. Finally, we use the element-wise sum to obtain the high-quality (HQ) output image $I_{high}$.

### 4.3 Low-Frequency State Space Block

The LFSSBlock is employed to extract and model low-frequency information flows from the spatial domain of feature embedding, as illustrated in Figure 2 right. Given the input low-frequency feature $F_L^i \in \mathbb{R}^{H \times W \times C}$, we initially apply Layer Normalization (LN), followed by the Vision State Space Module (VSSM), to capture the spatial long-term dependencies. Additionally, it incorporates a Gate Feed-Forward Network (GFFN) to improve the efficiency of channel information flow. This process can be formulated as follows:

$$Z = VSSM(LN(F_L^i)) + \beta \cdot F_L \tag{9}$$

$$F_L^{i+1} = GFFN(Z) + \gamma \cdot Z \tag{10}$$

where $VSSM(\cdot)$ and $GFFN(\cdot)$ denote VSSM and GFFN function, respectively. $LN(\cdot)$ denotes the operation of layer normalization. $Z$ denotes the intermediate hidden variable of LFSSBlock. $\beta$ and $\gamma$ represent the learnable scale factor.

**Vision State Space Module:** Inspired by the achievements of Mamba in modeling long-range dependencies with linear complexity, we have incorporated the VSSM into the UHD LLIE task. The VSSM can effectively model long-range dependencies with the state space equation. The architecture of VSSM is shown in Figure 2 right. Building upon prior work [18], the input feature $X \in \mathbb{R}^{H \times W \times C}$ is processed through two parallel branches. In the first branch, a linear layer expands the feature channel to $\lambda C$, with $\lambda$ being a predetermined channel expansion factor. This expansion is succeeded by

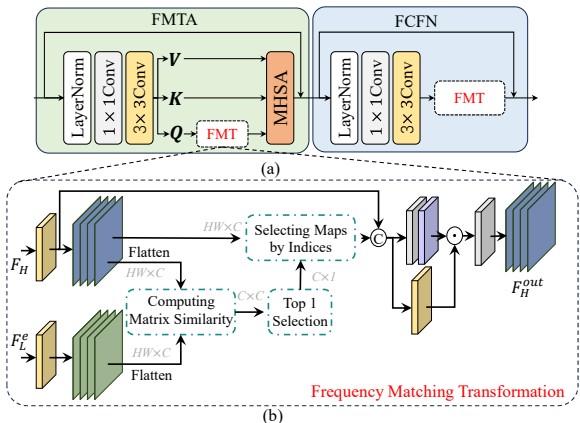

**Figure 5: (a) High-Frequency Enhance Block. (b) Frequency Matching Transformation.**

a depth-wise convolution and SiLU activation function, in conjunction with a 2D Selective Scan Module (2D-SSM) and LayerNorm. In the second branch, channel expansion to $\lambda C$ is achieved using a linear layer, then activated by SiLU. Following this, a Hadamard product combines the outputs of both branches. To finalize, channels are reduced back to $C$, producing an output $X_{out}$ same as the input dimensions. It can be written as follows:

$$X_1 = LN(\text{2D-SSM}(SiLU(DWConv(Linear(X))))) \tag{11}$$

$$X_2 = SiLU(Linear(X)) \tag{12}$$

$$X_{out} = Linear(X_1 \odot X_2) \tag{13}$$

where $DWConv(\cdot)$ and $Linear(\cdot)$ denote depth-wise convolution and linear projection. $\odot$ denotes the Hadamard product.

**Gate Feed-Forward Network**: In our framework, the Gated Feature Fusion Network (GFFN) employs a nonlinear gating mechanism to regulate information flow, enabling individual channels to concentrate on fine details that complement those from other layers. The operation of the GFFN is defined as follows:

$$F_{out} = W_p^3(\delta_{NG}(W_{d3}^2 W_p^2(LN(F_{in})))) \tag{14}$$

where $\delta_{NG}(\cdot)$ is the function of non-linear gate mechanism. Similar to SimpleGate [4], the non-linear gate mechanism divides the input along the channel dimension into two features $\mathbf{F}_1, \mathbf{F}_2 \in \mathcal{R}^{H \times W \times \frac{C}{2}}$. The output is then calculated by non-linear gating as $\delta_{NG}(\mathbf{F}') = GELU(\mathbf{F}_1) \odot \mathbf{F}_2$, where $GELU(\cdot)$ denotes the activation function. The $F_{in}$ and $F_{out}$ denote the input and output of GFFN.

**2D Selective Scan Module (2D-SSM):** The original Mamba processes input data causally, efficiently handling the sequential data typical of NLP tasks but encountering difficulties with non-sequential data like images. To adeptly manage 2D spatial information, we adopt the approach from [38] and deploy the 2D-SSM. As depicted in Figure 4, this technique transforms a 2D image feature into a linear sequence by scanning across four distinct orientations: top-left to bottom-right, bottom-right to top-left, top-right to bottom-left, and bottom-left to top-right. It then captures the extensive range dependencies for each sequence via the discrete state space equation. Subsequently, a summative merging of all sequences, followed by a reshaping process, reinstates the original 2D framework.

## 4.4 High-Frequency Enhance Block

To enhance the feature representation within the high-frequency components more effectively, we propose the construction of a feature transformation from the low-frequency component to the high-frequency component. This transformation aims to enhance the high-frequency components by leveraging similar information in the low-frequency component. To accomplish this goal, we introduce the High-Frequency Enhance Block (HFEBlock), as illustrated in Figure 5 (a). The HFEBlock enriches the missing information in high-frequency components by selecting more representative high-frequency similarity features in low-frequency components. This process is achieved through Frequency Matching Transformation (FMT), depicted in Figure 5 (b). Each HFEBlock contains a Frequency Matching Transformation Attention (FMTA) and a Frequency Correction Forward Network (FCFN), which are used to explore frequency matching and correction within the attention mechanism and forward network, respectively:

$$F'_H = FMTA(LN(F_H^{in}), F_L) + F_H^{in} \tag{15}$$

$$F_H^{out} = FCFN(LN(F'_H), F_L) + F' \tag{16}$$

where $F_H^{in}$ and $F_H^{out}$ denote the input and output high-frequency features of HFEBlock. $FMTA(\cdot, \cdot)$ and $FCFN(\cdot, \cdot)$ represent the operation of FMTA and FCFN, respectively.

**Frequency Matching Transformation Attention:** Drawing on previous work [23], the impact of a more potent query can be substantial on the outcomes. We enhance the query by substituting it with the refined and improved low-frequency features $F_L^e$, thereby imbuing the query with more significant content. We introduce the Feature-Modified Transformer Attention (FMTA) to achieve this enhancement of the query representation, thereby optimizing the attention mechanism. The FMTA initially produces the query (Q), key (K), and value (V) projections from the normalized high-frequency features $F_H^{in}$ through 1x1 convolution $W_p$ and 3x3 depthwise convolution $W_d$, followed by executing the FMT between Q and $F_L^e$. Subsequently, FMTA applies attention as follows:

$$FMTA(F_H^{in}, F_L^e) = \mathcal{A}(FMT(\mathbf{Q}, F_L^e), \mathbf{K}, \mathbf{V}) \tag{17}$$

where $\mathbf{Q, K, V} = Split(W_d W_p(F_H^{in}))$; $\mathcal{A}(\mathbf{Q,K,V}) = \mathbf{V} \cdot Softmax(\frac{\mathbf{K \cdot Q}}{\alpha})$. Here, $FMT(\cdot, \cdot)$, $Split(\cdot)$, and $Softmax(\cdot)$ mean the operation of FMT, split, and softmax. $\alpha$ is a learnable scaling parameter to control the magnitude of the dot product of K and Q.

**Frequency Correction Forward Network:** Initially, the Feed-Forward Network (FFN) is composed of a layer normalization, a 1x1 convolution, and a 3x3 depth-wise convolution. Similar to the FMTA, we incorporate the corresponding FMT within the FFN to facilitate enhanced augmentation of high-frequency component information. This process is represented as follows:

$$FCFN(F'_H, F_L^e) = FMT(W_d W_p(LN(F'_H)), F_L^e) \tag{18}$$

**Frequency Matching Transformation:** The FMT, depicted in Fig. 3(a), is designed to convert low-frequency features into high-frequency ones, thereby supplying the high-frequency component with more descriptive features through a correlation matching scheme to achieve superior enhancement. Initially, we compute a similarity matrix **M** between the high-frequency and low-frequency

components. Subsequently, we select the $Top-1$ vector $D$. The similar channels in the low-frequency component are then selected as the output based on the indices of vector $D$, which can be expressed as follows:

$$\mathbf{M} = Sim(F_L, F_H) \tag{19}$$

$$D = Top_1(M) \tag{20}$$

$$Y_{selected} = Select(F_L|Indices(D)) \tag{21}$$

where $Sim(\cdot, \cdot)$ denotes the operation of the computer similarity, which is measured in terms of Euclidean distances. $Select(\cdot|\cdot)$ and $Indices(\cdot)$ represent the select feature and obtain indices value operation.

Finally, the selected features $Y_{selected}$ are concatenated with the original high-frequency features, and a parallel branch is utilized to fuse the high-frequency features. Specifically, one branch calculates an attention map using a $1 \times 1$ convolution $W_p$ and a Sigmoid function. The other branch undergoes a $3 \times 3$ convolution $W_d$. The outputs of both branches are then multiplied and merged through another $1 \times 1$ convolution $W_p$ to produce the output features $F_H^{out}$.

$$Y_{selected}^{concat} = Concat(Y_{selected}, F_H) \tag{22}$$

$$F_H^{out} = Wp(Sigmoid(Wp(Y_{selected}^{concat})) \odot W_d(Y_{selected}^{concat})) \tag{23}$$

## 5 EXPERIMENTS

### 5.1 Implementation Details

The number of LFSSBlocks is $[1, 2, 4]$ and the number of HFEBlocks is $[1, 1, 1]$ at each layer in the network encoder and decoder. The number of attention heads is 8, and the number of channels $C$ is 32. We train models using AdamW [19] optimizer with the initial learning rate $5 \times 10^{-4}$ gradually reduced to $1 \times 10^{-7}$ with the cosine annealing for a total 500k iterations. We use random rotations of 90, 180, 270, random flips, and random cropping to $512 \times 512$ size for the augmented training data. To constrain the training of Wave-Mamba, we use the $L_1$ loss function. All experiments are conducted on two NVIDIA 3090 GPUs.

### 5.2 Datasets

Our experiments leverage three prominent benchmarks for evaluating low-light image enhancement algorithms:

The UHD-LL dataset [23] is a real-world, paired image collection that contains 2,150 pairs of 4K ultra-high-definition (UHD) data saved in 8-bit sRGB format. This dataset is split into 2,000 training pairs and 115 test pairs.

The UHD-LOL benchmark [13] consists of two subsets - UHD-LOL4K and UHD-LOL8K - featuring 4K and 8K resolution images captured under low-light conditions, respectively. For our study, we utilize the UHD-LOL4K subset, which encompasses a total of 8,099 image pairs. Of these, 5,999 pairs are allocated for training, and the remaining 2,100 pairs are reserved for testing.

In addition, we also evaluate our method on the widely adopted LOL dataset [28], which is a standard benchmark for low-light image enhancement algorithms. The LOL dataset contains 500 image pairs in total, with 485 pairs used for training and 15 pairs set aside for testing. We also used LOLv2-Real to test model performance.

**Table 1: Comparison of quantitative results on UHD-LOL4K and UHD-LL datasets. The best and second best values are indicated with bold text and underlined text respectively.**

| Type | Method | Venue | UHD-LOL4K PSNR↑ | SSIM↑ | LPIPS↓ | UHD-LL PSNR↑ | SSIM↑ | LPIPS↓ | Average PSNR↑ | SSIM↑ | LPIPS↓ | Parameter↓ |
|---|---|---|---|---|---|---|---|---|---|---|---|---|
| non-UHD | Zero-DCE [10] | CVPR'20 | 17.19 | 0.850 | 0.193 | 17.08 | 0.663 | 0.513 | 17.14 | 0.757 | 0.353 | 79.416K |
| | Zero-DCE++ [14] | TPAMI'21 | 15.58 | 0.835 | 0.222 | 16.41 | 0.630 | 0.530 | 16.00 | 0.733 | 0.376 | 10.561K |
| | RUAS [17] | CVPR'22 | 14.68 | 0.758 | 0.274 | 13.56 | 0.749 | 0.460 | 14.12 | 0.754 | 0.367 | **3.438K** |
| | Uformer [27] | CVPR'22 | 29.99 | 0.980 | 0.034 | 19.28 | 0.849 | 0.356 | 24.64 | 0.915 | 0.195 | 20.628M |
| | Restormer [35] | CVPR'22 | 36.91 | 0.988 | 0.023 | 22.25 | 0.871 | 0.289 | 29.58 | 0.930 | 0.156 | 26.112M |
| | DiffLL [12] | Siggraph'23 | 36.95 | 0.989 | 0.022 | 21.36 | 0.872 | 0.239 | 29.15 | 0.930 | 0.131 | 17.29M |
| UHD | LLFormer [13] | AAAI'23 | 37.33 | 0.989 | 0.020 | 22.79 | 0.853 | 0.264 | 30.06 | 0.921 | 0.142 | 13.152M |
| | UHDFour [15] | ICLR'23 | 36.12 | **0.990** | 0.021 | 26.22 | 0.900 | 0.239 | 31.17 | 0.945 | 0.130 | 17.537M |
| | UHDFormer [23] | AAAI'24 | 36.28 | 0.989 | 0.020 | 27.11 | **0.927** | 0.245 | 31.69 | **0.958** | 0.130 | 339.3K |
| | Wave-Mamba (Ours) | - | **37.43** | **0.990** | **0.019** | **27.35** | 0.913 | **0.185** | **32.39** | 0.952 | **0.102** | 1.258M |

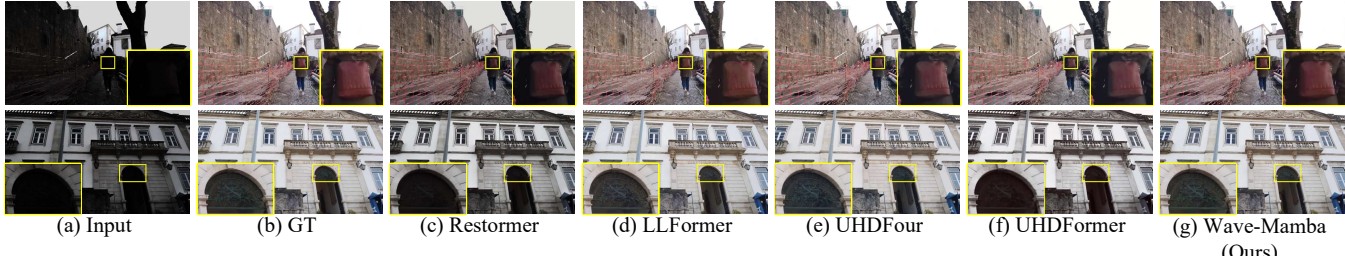

(a) Input    (b) GT    (c) Restormer    (d) LLFormer    (e) UHDFour    (f) UHDFormer    (g) Wave-Mamba (Ours)

**Figure 6: Visual comparison results on the UHDLOL4K dataset. The proposed method produces visually more pleasing results. (Zoom in for the best view)**

**Table 2: Comparison of quantitative results on LOLv1 and LOLv2-Real datasets. The best and second best values are indicated with bold text and underlined text respectively.**

| Type | Method | LOLv1 PSNR | SSIM | LOLv2-Real PSNR | SSIM |
|---|---|---|---|---|---|
| non-UHD | Uformer [27] | 18.55 | 0.721 | 18.44 | 0.759 |
| | Restormer [35] | 22.37 | 0.816 | 24.91 | 0.851 |
| | Retinexformer [2] | 25.16 | 0.845 | 22.80 | 0.840 |
| | MambaIR [11] | 22.31 | 0.826 | 21.25 | 0.831 |
| | RetinexMamba [1] | 24.03 | 0.827 | 22.45 | 0.844 |
| | MambaLLIE [30] | - | - | 22.95 | 0.847 |
| | DiffLL [12] | 26.34 | 0.845 | 28.86 | 0.876 |
| UHD | LLFormer [13] | 23.65 | 0.816 | 27.75 | 0.861 |
| | UHDFour [15] | 23.09 | 0.871 | 21.78 | 0.870 |
| | Wave-Mamba (Ours) | **26.54** | **0.883** | **29.04** | **0.901** |

## 5.3 Comparisons with State-of-the-Art Methods

In this section, we compare our proposed Wave-Mamba quantitatively and qualitatively with the current State-of-the-Art methods. We use the PSNR, SSIM [26], and LPIPS [37] to evaluate our method. PSNR measures the quality of reconstructed images by comparing pixel intensity differences, SSIM evaluates image quality based on luminance, contrast, and structure, and LPIPS assesses perceptual similarity using deep learning models.

**Quantitative Results:** To validate the effectiveness of our method on the UHD dataset, we compared it with state-of-the-art (SOTA) low-light enhancement methods, including Zero-DCE [10], Zero-DCE++ [14], RUAS [17], Uformer [27], Restormer [35], LLFormer [13], UHDFour [15], and UHDFormer [23]. Given that the vast majority of existing methods are not designed to handle the full resolution of UHD images, we employ a sliding window approach to

generate the final enhanced image. This way involves chunking the input UHD image into patches, inference each patch independently, and then stitching the predictions back together to obtain the output. Table 1 shows the performance of our Wave-Mamba and other methods. As shown in Table 1, our approach achieves the best PSNR performance results with few parameters on all UHD datasets. In addition to this, our method is also far better in terms of perceptual metrics than the current most superior UHDFormer.

For further validation of the effectiveness of our methods, we additionally compare these methods on the LOLv1 and LOLv2-Real datasets, as shown in Table 2. Due to the low-resolution images in the LOLv1 and LOLv2-Real datasets, the downsampling operation in the existing UHD LLIE methods can cause information loss, which leads to poor results. In contrast, benefiting from the information preservation feature of the wavelet transform, our method can also achieve state-of-the-art results on low-resolution datasets. All these results clearly suggest the outstanding effectiveness and efficiency advantage of our Wave-Mamba.

**Qualitative Results:** To complement the quantitative assessments, we also conduct qualitative comparisons of our proposed method. As shown in Figures 6 and 7, existing UHD LLIE methods that rely on high-magnification downsampling suffer from the issue of introducing artifacts in the restored images. Furthermore, in comparison to the state-of-the-art UHDFormer approach, our Wave-Mamba method can achieve results that are closer to the true colors and textural details. Our Wave-Mamba demonstrates its capabilities in enhancing low-visibility and low-contrast regions, reliably removing noise without introducing unwanted artifacts, and robustly preserving the original color information. More results are provided in the supplementary material.

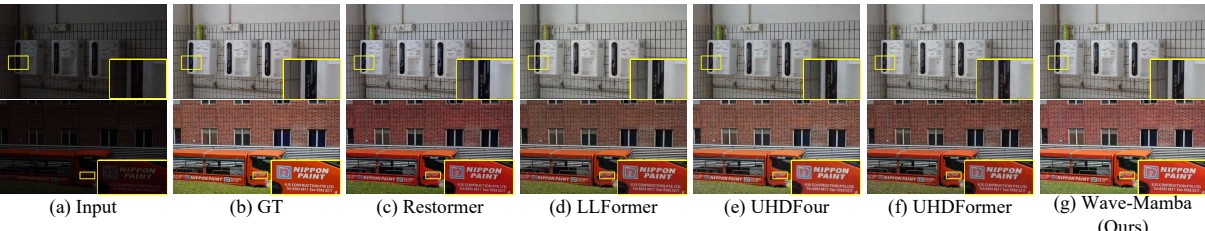

| (a) Input | (b) GT | (c) Restormer | (d) LLFormer | (e) UHDFour | (f) UHDFormer | (g) Wave-Mamba (Ours) |

**Figure 7: Visual comparison results on the UHDLL dataset. The proposed method produces visually more pleasing results. (Zoom in for the best view)**

**Table 3: Ablation studies of different components.**

| Experiment | LFSSBlock | HFEBlock | FMT | PSNR | SSIM |
|---|---|---|---|---|---|
| Setting1 | | ✔ | | 10.14 | 0.301 |
| Setting2 | | ✔ | ✔ | 12.35 | 0.346 |
| Setting3 | Residual Block | | | 24.66 | 0.845 |
| Setting4 | ✔ | | | 25.94 | 0.887 |
| Setting5 | ✔ | | ✔ | 26.41 | 0.903 |
| Setting6 | ✔ | ✔ | | 27.13 | 0.908 |
| Full Model | ✔ | ✔ | ✔ | 27.35 | 0.913 |

**Table 4: Ablation studies on different numbers of LFSSBlocks and HFEBlocks.**

| Experiment | LFSSBlocks | HFEBlocks | PSNR | SSIM | Params |
|---|---|---|---|---|---|
| 1 | [1,1,4] | [1,1,1] | 26.48 | 0.904 | 1.2M |
| 2 | [1,2,4] | [1,1,1] | 27.35 | 0.913 | 1.3M |
| 3 | [1,1,4] | [1,1,2] | 26.67 | 0.909 | 2.2M |
| 4 | [1,2,4] | [1,1,2] | 27.41 | 0.922 | 2.3M |
| 5 | [1,1,2,4] | [1,1,1,1] | 27.38 | 0.913 | 2.3M |

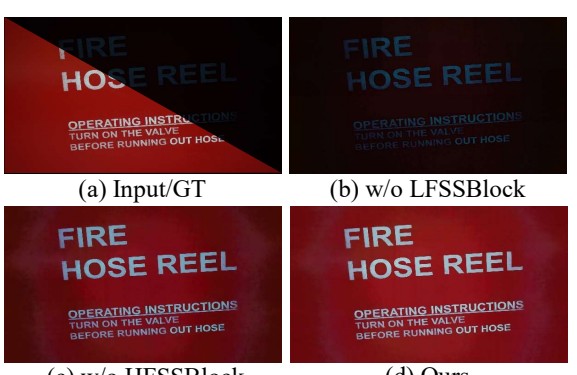

| (a) Input/GT | (b) w/o LFSSBlock |
| (c) w/o HFSSBlock | (d) Ours |

**Figure 8: Visual effect on our proposed blocks. (Zoom in for the best view)**

## 5.4 Ablation Study

In this section, we utilize the UHD-LL dataset to perform an ablation study evaluating the key design choices in our Wave-Mamba model. More results are provided in the supplementary material.

**Effectiveness of Proposed Blocks:** We present ablation studies to demonstrate the effectiveness of the main component in our design, including LFSSBlock, HFEBlock, and FMT. The experimental results are shown in Table 3. As shown in the results of Setting 1 and 2 in Table 3, the lack of restoration and enhancement of low-frequency information is unable to improve image quality. Furthermore, we conducted a comparative experiment using

Residual Blocks and LFSSBlocks with similar parameters, which demonstrates that our proposed LFSSBlock, leveraging its powerful global modeling capability through SSMs, can achieve better performance. Additionally, the ablation study on the HFEBlock indicates that while high-frequency information cannot primarily determine the quality of the restored image, it does have some influence on the restoration results. Therefore, compared to simply stacking high-frequency information, the addition of the HFEBlock with FMT in our method allows us to achieve optimal performance. The visualization in Figure 8 also shows that our proposed method recovers clearer images. The above experiments fully demonstrate the effectiveness of the modules we proposed.

**Tradeoff Study of Performance Versus Parameters:** To find out the influence of the computation resources, we explore different numbers of LFSSBlocks and HFEBlocks to construct network structures of different depths. The detailed experimental results are shown in the Table 4. As can be seen in Table 4, the HFEBlocks have a greater impact on the parameters of the model compared to the LFSSBlocks, but the performance improvement is relatively small. Furthermore, during the process of increasing the number of LFSS-Blocks, we found that increasing the number of the second-level LFSSBlocks leads to a greater performance boost than increasing the number of the last-level LFSSBlocks. To verify the impact of increasing the second-level LFSSBlocks on performance, we compared Experiments 2 and 3, and found that performance did not increase as the number of modules grew. Considering the balance between performance and computational cost, we adopted the configuration of Experiment 2 as our final setting.

## 6 CONCLUSION

In this paper, inspired by the characteristics of real low-light images in the wavelet domain, we propose a novel paradigm called Wave-Mamba that leverages state space models (SSMs) for the UHD LLIE task. Specifically, we employ wavelet-based downsampling to avoid the information loss issues associated with high-magnification downsampling. Additionally, we introduce Mamba to propose a low-frequency state space block (LFSSBlock), which relies on the excellent global modeling capability of SSMs to obtain excellent recovery results. Furthermore, to exploit the similarities between different frequency components, we design a high-frequency enhance block (HFEBlock) that utilizes the low-frequency information to correct the high-frequency details. Owing to these unique designs targeting the different frequency components in the wavelet domain, our Wave-Mamba outperforms state-of-the-art methods in UHD LLIE with appealing efficiency.

# ACKNOWLEDGMENTS

This work was funded by Science and Technology Project of Guangzhou (202103010003), Science and Technology Project in key areas of Foshan (2020001006285), Xijiang Innovation Team Project (XJCXTD3-2019-04B).

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
