# OpenReview forum: "Wave-Mamba: Wavelet State Space Model for Ultra-High-Definition Low-Light Image Enhancement"
_acmmm.org/ACMMM/2024/Conference — MM2024 Poster_

### Official Review · Reviewer_tWub · 2024-05-13

**Rating:** 3
**Confidence:** 3

**Summary:**

This paper proposes a novel approach that combines wavelet transform and state-space models (SSMs) for ultra-high-definition (UHD) low-light image enhancement (LLIE). This method uses  a low-frequency state space (LFSS) module by improving SSMs to focus on restoring the information of low-frequency sub-bands, and proposes a high-frequency enhancement module (HFEM) for high-frequency sub-band information. Numerous qualitative and quantitative experiments have demonstrated the effectiveness and efficiency of the proposed method.

**Strengths:**

The proposed method has sufficient detailed background introduction, novel insights, and clear motivation. This paper provides a detailed introduction to the background of LLIE and the shortcomings of UHD LLIE. At the same time, the motivation of the paper is clear, cleverly combining the characteristics of wavelet transform and mamba for UHD LLIE tasks.

**Limitations:**

(1) Line 132, "Therefore, Therefore, ......", There is an incorrect spelling here.
(2) Section 3.1 is empty.
(3) Line 440, "In line with the observations made in Sec. 3.2, ......", Section 3.2 does not exist.
(4) Lines 460-461, " by Selective Kernel Feature Fusion (SKFF) []", The reference is empty.
(5) Why was UHD-LOL8K not selected in Section 5.2.
(6) Need to add references to estimated metrics, SSIM and LPIPS, and provide a brief introduction to them.
(7) Not mandatory. Can we add a set of experiments in Table 4 with parameters set to LFSSBlocks: [1,1,4], HFEBlocks: [1,1,2].

Overall, I think the overall quality of this paper is good, but there were significant errors in the writing. Therefore, currently I tend to reject this paper. The author should reshape the content of Sections 3 and 4

**Suitability:**

3

---

### Official Review · Reviewer_Zk96 · 2024-05-13

**Rating:** 4
**Confidence:** 3

**Summary:**

The authors propose a wavelet-based Mamba (Wave-Mamba) for ultra-high-definition low-light image enhancement. The Wave-Mamba is based on two pivotal insights derived from the wavelet domain: 1) most of the content information of an image exists in the low-frequency component, less in the high-frequency component. 2) The high-frequency component exerts a minimal influence on the outcomes of low-light enhancement. The Wave-Mamba is composed of low-frequency state space (LFSS) modules and high-frequency enhancement modules (HFEMs) to leverage the advantages of both wavelet transform and Mamba. The experimental results seem good.

**Strengths:**

1) The authors are the pioneers in introducing Mamba to UHD LLIE task.
2) By adopting Mamba to wavelet domain, Wave-Mamba exploits the wavelet transform to avoid information loss, and to overcome the shortcomings of SSMs which are insensitive to noise.
3) The authors propose two novel module LFSS-block and HFE-block in Wave-Mamba.
4) The experimental results are promising compared with recent low-light image enhancement methods.

**Limitations:**

1) In the observations, the wavelet transform is adopted in image domain. However, the wavelet transform is adopted in the feature domain after convolutional layers in Wave-Mamba. The inconsistency between these two domains should be discussed.
2) The scales of wavelet transform should be ablation studied. Why do the authors choose to adopt four times wavelet transform?
3) In eq.(19-21), it is confusing how to select the Top-1 vector D and how to select low-frequency components on the indices of D? F_L is of size HWxC, D is of size 1xC. What is the size of Y_selected?
4) Lots of typos in the manuscript, for example:
a) on page 2, left column, “Therefor, Therefore, how to…”;
b) on page 3, right column, “As illustrated in Figure 1” should be Figure 2 according to context;
c) on page 4, right column, “(SKFE)[]”.

**Suitability:**

2

---

### Official Review · Reviewer_qgVT · 2024-05-24

**Rating:** 4
**Confidence:** 4

**Summary:**

Based on the observation of the wavelet domain, this paper proposes a method that combines wavelet transform and state-space models to address the high computational complexity and information loss in UHD LLIE. Moreover, a high-frequency enhancement module is introduced to restore the high-frequency details. Experimental results demonstrate the effectiveness of the method on UHD LLIE.

**Strengths:**

1. The motivation of this paper is reasonable, and there is sufficient discussion and validation to support the motivation.
2. There are numerous experiments, demonstrating the effectiveness of the method through quantitative metrics and visual results

**Limitations:**

1. The efficiency of the method encompasses not only the number of parameters but also requires comparisons of computational load and runtime. I will consider raising the rating if the author can provide comparisons of computational load and runtime with other approaches.
2. Some typographical errors in the paper need to be modified, such as those on line 132 and line 298.

**Suitability:**

3

---

### Official Review · Reviewer_ACFL · 2024-05-25

**Rating:** 4
**Confidence:** 2

**Summary:**

This paper introduces wavelet transform and Mamba architecture into UHD low-light image enhancement. In detail, the authors design encoder-decoder architecture and consider low-/high-frequency information separately. Experimental results prove the effectiveness of their proposed method in UHD LLIE.

**Strengths:**

Overall, this paper is well-written and motivation is clear.

**Limitations:**

- Lack of comparison with existing Mamba-based restoration model.
- Previous ACMMM papers on Low-Light Image Enhancement (LLIE) [1-2] should be discussed and cited to establish the relevance of this paper to ACMMM.

[1] "Learnability Enhancement for Low-light Raw Denoising: Where Paired Real Data Meets Noise Modeling," ACMMM 2022.

[2] "FourLLIE: Boosting Low-Light Image Enhancement by Fourier Frequency Information," ACMMM 2023.

**Suitability:**

3

---

### Meta-Review · Area_Chair_YzPP · 2024-07-05

**Recommendation:** Accept (Poster)
**Confidence:** 3

**Metareview:**

The authors propose a method that uses combines wavelet transform and state-space models (SSMs) for ultra-high-definition (UHD) low-light image enhancement (LLIE). The method is technically sound and sufficient novel for publication. It is clear and the authors have performed enough experimental validation. In their rebuttal, the authors have addressed issues concerning the comparison with other methods and complexity.